# What Makes Bangladeshi Pregnant Women More Compliant to Iron–Folic Acid Supplementation: A Nationally Representative Cross-Sectional Survey Result

**DOI:** 10.3390/nu15061512

**Published:** 2023-03-21

**Authors:** Kazi Istiaque Sanin, Mahbubul Alam Shaun, Razia Sultana Rita, Md. Khaledul Hasan, Mansura Khanam, Md. Ahshanul Haque

**Affiliations:** 1Nutrition and Clinical Services Division, International Centre for Diarrhoeal Diseases Research, Bangladesh, 68 Shaheed Tajuddin Ahmed Sarani, Mohakhali, Dhaka 1212, Bangladesh; 2Department of Biochemistry and Food Analysis, Faculty of Nutrition and Food Science, Patuakhali Science and Technology University, Dumki, Barisal 8602, Bangladesh

**Keywords:** iron–folic acid supplementation, IFA compliance, pregnancy, anemia, Bangladesh

## Abstract

Background: Iron–Folic Acid Supplementation (IFAS) is an effective strategy to prevent iron deficiency anemia during pregnancy. We aimed to explore the key factors associated with compliance to IFA tablets in Bangladesh. Methods: This study analyzed the 2017–2018 Bangladesh Demographic and Health Survey data of 3828 pregnant women aged 15–49 years. We categorized compliance into two categories; at least 90 days and full 180 days of consumption. We performed multivariable logistic regression to identify the association between key factors and IFAS compliance. Results: The prevalence of consumption of IFA tablets for at least 90 days was 60.64%, and only 21.72% of women consumed the IFA for the full 180 days. About three-quarters of the women (73.36%) having at least four antenatal care visits (ANC) consumed IFA for at least 90 days, whereas only three in ten women (30.37%) consumed IFA for a minimum of 180 days. For compliance with IFA for at least 90 days, respondent’s age of 20–34 years (aOR 1.26, 95% CI 1.03–1.54), respondent’s educational qualification of secondary (aOR 1.77, 95% CI 1.16–2.70) or higher (aOR 2.73, 95% CI 1.65–4.53), husband’s educational qualification of secondary (aOR 1.33, 95% CI 1.00–1.77) or higher (aOR 1.75, 95% CI 1.22–2.52), and having received at least four antenatal care (ANC) visits from medically skilled providers (aOR 2.53, 95% CI 2.14–3.00) were significantly associated with higher odds of compliance. For compliance with IFA for at least 180 days, respondent’s educational qualification of higher (aOR 2.45, 95% CI 1.34–4.48), and having received at least four ANC visits from medically skilled providers (aOR 2.43, 95% CI 1.97–3.00) were significantly associated with higher odds of compliance. Intimate partner violence was negatively associated with compliance with IFA for at least 180 days (aOR 0.62, 95% CI 0.48–0.81). Conclusions: The full compliance to IFAS is still sub-optimal in Bangladesh. Further precise context-specific intervention strategies must be developed and implemented with fidelity.

## 1. Introduction

Pregnancy is one of the most crucial periods, requiring additional nutritional demands, predominantly micronutrients, to maintain the physiological adaptation of the mother as well as the metabolic needs of the embryo/fetus [1]. Among the vital micronutrients, women are more vulnerable to anemia during pregnancy due to iron deficiency, which can be preventable by consuming iron–folic acid supplements [2]. Globally, over 40% of pregnant women are anemic, of which 50% are caused by iron deficiency [3], whereas in Bangladesh, it stands at 42% among pregnant women [4]. Anemia leads to maternal deaths and enhances the risks of hemorrhage, rupture of membranes, and reduced labor capacity among women. Moreover, it increases the chances of adverse birth outcomes in the form of low birth weight, impaired cognitive development, stillbirths, small for gestational age, and premature birth, which are major causes of neonatal mortality in developing nations [5,6,7,8]. To alleviate the status of iron deficiency and adverse pregnancy outcomes, the World Health Organization (WHO) recommends the consumption of iron–folic acid supplements (IFA) (30–60 mg of iron and 400 μg of folic acid) for a minimum of 180 days [9]. As per policy, IFA supplementation in Bangladesh is aimed to be delivered through antenatal checkups (ANC) that function within the government healthcare system. However, studies have shown that only about 26–36% of Bangladeshi pregnant women consume the recommended IFA dosage [10,11]. A cross-sectional study using the Bangladesh Demographic and Health Survey (BDHS) 2004, 2007, and 2011 reported that IFA was delivered to only 32.8% of women [12]. Another study in Bangladesh found that out of 180 recommended tablets, participants only took an average of 94 tablets [13]. 

Compliance with IFA among pregnant women depends on numerous factors that regulate adherence to IFA [9,14]. Independent factors associated with IFA compliance found in former studies were the age of women [5,15,16,17], education of pregnant mothers [5,15], partners’ academic qualifications [5], household’s economic status [5,17], number of antenatal care visits [5,13,18,19], self-efficiency or working status [16,20], knowledge on anemia [19], counseling during ANC visits [20], women’s decision-making capabilities [21] and intimate partner violence (IPV) [22]. Recently, several pieces of literature have found that IPV has become a major obstacle to the utilization of maternal healthcare services during the antenatal period, particularly adherence to IFAS during pregnancy [22,23,24,25]. However, in Bangladesh, no studies have yet been conducted focusing on the impact of IPV on compliance with IFA.

Despite being freely distributed as per the government’s mandate, the low compliance with IFA consumption among Bangladeshi pregnant women calls for further analysis. Several studies have identified the predictors of IFA compliance [26,27,28]. Our study aims to determine the association between the established factors and compliance with IFA and to identify potential role of unexplored factors such as IPV among Bangladeshi pregnant women. 

## 2. Materials and Methods

### 2.1. Source of Data and Sample Size

This study was based on a secondary dataset of the Bangladesh Demographic and Health Survey 2017–2018 (BDHS 2017–18), a nationally representative cross-sectional survey [29]. Data were collected using two-stage stratified household sampling in which six types of questionnaires were used: (1) the Household Questionnaire, (2) the Woman’s Questionnaire (completed by ever-married women aged 15–49), (3) the Biomarker Questionnaire, (4) two verbal autopsy questionnaires to collect data on causes of death among children under the age of 5, (5) the Community Questionnaire, and (6) the Fieldworker Questionnaire. For our analysis, we used the women’s data file (BDIR7RFL), which had a total sample size of 20,127 women aged 15–49 years. However, based on our objective, we only extracted information from those women who had ever consumed at least one IFA during their pregnancy. After cleaning missing data, our study’s final (weighted) sample size was 3828. The details regarding sampling weight are given in the analysis section. 

### 2.2. Iron–Folic Acid Supplementation Compliance Assessment

In the BDHS survey, women were asked to report the number of days they took iron tablets or syrup in response to a question on their consumption of IFA. The prevalence of consumption for a minimum of 90 days and a minimum of 180 days was calculated based on the self-reported intake to assess compliance with the World Health Organization (WHO) recommendation of at least six months of iron and folic acid (IFA) supplementation [30,31]. Optimal compliance was defined as the consumption of IFA for at least 90 days, while full compliance was defined as the consumption of IFA for at least six months, according to previous studies [9]. The number of days of consumption of IFA for at least 90 days and 180 days were treated as two separate outcome variables in this study. By using these outcome variables, the study aimed to investigate the prevalence of IFA supplementation compliance among women and identify factors associated with compliance.

### 2.3. Independent Variables

Several socioeconomic and demographic variables of the respondents were considered independent variables based on previous studies [5,13,15,16,17,18,19,20,21,22]. These were women’s age, respondents and their husband’s educational level, wealth index of household, current working status of participants, at least four ANC visits from medically skilled providers, decision making by respondents, mass media exposure by respondents, and intimate partner violence. Wealth index was constructed using principal component analysis based on several socioeconomic variables such as household assets and material, toilet facilities, source of drinking water, and fuel consumption. The index was created using the first component and then divided into five quintiles, ranging from the poorest to the richest households. However, in our analysis, we combined the first and last two quintiles to create three groups: poor, middle, and rich. The variable “at least four ANC visits from medically skilled providers” was used to measure the level of antenatal care received by women during their last pregnancy. The variable was defined as a woman having received antenatal care four or more times, with each visit being provided by a skilled service provider. The skilled service providers considered in this study included MBBS doctors, nurse/midwife/paramedic, family welfare visitor, community skilled birth attendant, and sub-assistant community medical officer. A decision-making variable was created based on three indicators: the woman’s ability to make decisions on major household purchases, healthcare, and visits to family or relatives, either independently or jointly with her husband. The variable was then converted to a binary form if the woman responded positively to all three indicators. Intimate partner violence (IPV) has been defined in the Bangladesh Demographic and Health Survey as any behavior by a husband or partner that causes physical or emotional harm, including slapping, punching, kicking, or any type of physical violence, as well as forcing the wife or partner to perform sexual acts against her will, and emotional or mental abuse. The other two geographic variables were administrative division and place of residence. The division of Bangladesh refers to the country’s first-level administrative division and is divided into a total of eight regions: Barishal, Chattogram, Dhaka, Khulna, Mymensingh, Rajshahi, Rangpur, and Sylhet. Based on residence type, all the divisions are a mix of both urban and rural areas. Urban areas in Bangladesh are characterized by high population density, modern infrastructure, and a more diverse economy. They are also home to various industries, services, and institutions. Conversely, rural areas in Bangladesh are characterized by low population density, an agriculture-based economy, and traditional lifestyles. The majority of the population in Bangladesh live in rural areas, and agriculture is the main source of income for most people.

### 2.4. Sampling Weight

Due to the nationally representative data used in this study, the purpose of weighting the samples during statistical analysis was to account for the complex survey design and ensure that the survey results accurately represent the population from which the sample was drawn. Sampling weights were calculated for each individual in the sample and were adjusted for any unequal probabilities of selection across the sample strata and clusters, reflecting the probability of selection and any non-response bias. Specifically, the BDHS sampling weight was calculated as the inverse of the probability of selection of the individual or household. It was crucial to use sampling weights in all analyses of BDHS data to obtain unbiased estimates and standard errors and ensure reliable and accurate results.

### 2.5. Statistical Analysis

We performed statistical analyses to investigate the factors associated with IFA compliance. All statistical analyses were conducted using Stata 14.0 software and followed the guidelines of the Guide to DHS Statistics for Data Analysis [32]. The statistical methods used in this manuscript were designed to explore the associations between various independent variables and two outcome variables related to the consumption of IFA (iron and folic acid) for at least 90 and 180 days. To visualize the outcome variables in terms of intimate partner violence (IPV), bar diagrams were used. All independent variables were categorical and were summarized using frequencies and percentages, with data segregated by residence type. To assess the bivariate associations between the independent variables and the two outcome variables, a chi-squared test was performed using cross-tabulation. Simple logistic regression was then used to estimate the odds ratio (OR) as a measure of the strength of association. Finally, multiple logistic regression was used to estimate the adjusted OR (aOR) as the adjusted strength of association between compliance with IFA tablets for at least 90 days and 180 days and their associated factors. The relevant independent variables were included in the multiple models based on the bivariate analyses and a review of the relevant literature. The analyses were conducted using nationally representative data, with sampling weights used in all analyses [29]. The significance level was set at *p* < 0.05, with a 95% confidence interval (CI) used to assess the strength of the effect.

## 3. Results

### 3.1. Background Characteristics of Respondents 

Table 1 provides background characteristics of the study population, which is divided into urban and rural areas. The table includes information on division, respondent’s age, educational qualification of respondent and her husband, household’s wealth index, respondent’s working status, household head’s sex, receiving at least four ANC visits from medically skilled providers, decision making by respondents, access to mass media, and intimate partner violence. The table shows that the majority of the study population reside in rural areas (71.8%). In terms of division, the highest proportion of respondents live in Dhaka (25.8%), followed by Chattogram (21.0%). The majority of respondents are in the age group of 20–34 years (75.5%). A large proportion of respondents have secondary education (49.9%), while a significant proportion of husbands have primary education (31.8%). In terms of wealth index, the majority of households belong to the rich category (43.2%). In urban areas, 72.01% of respondents reported currently working, while in rural areas, 59.66% of respondents reported the same. In terms of household head’s sex, 90.9% of urban respondents reported having a male household head, while 86.25% of rural respondents reported the same. About two-thirds of the respondents received at least four ANC from medically skilled providers (51.7%), and about half of the respondents reported being involved in decision making (54.6%). Access to mass media is higher in urban areas (82.7%) compared to rural areas (64.2%). The prevalence of intimate partner violence is 17.4% in the study population.

### 3.2. Status of IFA Compliance 

Figure 1 presents data on the distribution of compliance rates with IFA (iron and folic acid) tablets for at least 90 days and 180 days. Along with overall compliance, the data are further broken down by whether or not the participants reported experiencing IPV. The overall compliance rate with IFA tablets for at least 90 days was 60.64%, with 54.09% of those who reported experiencing IPV being compliant and 62.02% of those who did not report experiencing IPV being compliant. The overall compliance rate with IFA tablets for at least 180 days was lower at 21.72%, with only 13.41% of those who reported experiencing IPV being compliant and 23.47% of those who did not report experiencing IPV being compliant. These findings describe that there was a lower level of compliance with IFA tablets for at least 180 days among women who have reported experiencing intimate partner violence, compared to those who have not reported such experiences.

### 3.3. Association between Key Factors and IFAS Compliance 

Table 2 presents the results of a multiple logistic regression analysis of factors associated with compliance with iron–folic acid (IFA) supplementation for at least 90 days and 180 days during pregnancy. The odds ratios (ORs) and 95% confidence intervals (CIs) are presented for each independent variable, along with the *p*-value indicating the statistical significance of the association.

For compliance with IFA for at least 90 days, the following variables were significantly associated with higher odds of compliance: respondent’s age of 20–34 years (aOR 1.26, 95% CI 1.03–1.54), respondent’s educational qualification of secondary (aOR 1.77, 95% CI 1.16–2.70) or higher (aOR 2.73, 95% CI 1.65–4.53), husband’s educational qualification of secondary (aOR 1.33, 95% CI 1.00–1.77) or higher (aOR 1.75, 95% CI 1.22–2.52), and having received at least four antenatal care (ANC) visits from medically skilled providers (aOR 2.53, 95% CI 2.14–3.00). Not working during pregnancy was not significantly associated with compliance with IFA for at least 90 days.

For compliance with IFA for at least 180 days, the following variables were significantly associated with higher odds of compliance: respondent’s age of 35–49 years (aOR 1.51, 95% CI 0.99–2.31, *p* = 0.057, borderline significance), respondent’s educational qualification of higher (aOR 2.45, 95% CI 1.34–4.48), husband’s educational qualification of higher (aOR 1.36, 95% CI 0.89–2.10, *p* = 0.156, not significant), not working during pregnancy (aOR 1.24, 95% CI 1.01–1.52), and having received at least four ANC visits from medically skilled providers (aOR 2.43, 95% CI 1.97–3.00). Intimate partner violence was negatively associated with compliance with IFA for at least 180 days (aOR 0.62, 95% CI 0.48–0.81).

The results suggest that younger age, lower educational qualification, and not receiving enough ANC visits were associated with lower compliance with IFA during pregnancy. Having a higher level of education, receiving enough ANC visits, and not experiencing intimate partner violence were associated with higher compliance with IFA during pregnancy. Additionally, not working during pregnancy was associated with higher compliance with IFA for at least 180 days.

## 4. Discussion

Iron folic acid supplementation during pregnancy is a well-established intervention endorsed by WHO and being delivered through the government healthcare system free of cost for decades in Bangladesh. Based on the latest Bangladesh Demographic and Health Survey data, we aimed to explore the current status of IFA consumption and investigate the associated factors among Bangladeshi pregnant women. We found that about 61% of Bangladeshi women consumed IFA tablets for at least 90 days, whereas the percentage of women consuming 180 tablets was only around 22%. Respondent’s age, respondent’s educational qualification, husband’s educational qualification, and receiving at least four antenatal care (ANC) visits from medically skilled providers were significantly and positively associated with consumption of IFA tablets for at least 90 days. Respondent’s educational qualification, receiving at least four ANC visits from medically skilled providers were significantly and positively associated with consumption of IFA for at least 180 days, whereas intimate partner violence was negatively associated.

Studies have shown that pregnant women in LMICs have a lower compliance with iron–folic acid (IFA) supplementation compared to women in high-income countries, despite the widespread distribution of IFA supplements in LMICs [14,33,34]. Low compliance has been attributed to several factors, including poor knowledge about the importance of IFA, fear of side effects, forgetfulness, lack of access, and inadequate counseling by health workers. In addition, sociocultural beliefs and practices, such as taboos and restrictions on food and drink during pregnancy, can also affect compliance with IFA supplementation [5,35,36,37]. A study conducted in Hawassa city, South Ethiopia, found that only 27.9% of pregnant women adhered to IFA supplementation [33]. Similarly, a study conducted in Aykel town, Northwest Ethiopia, reported a low adherence rate of 36.2% [34]. A study conducted in Southern Italy [35] found that 60.8% of pregnant women reported taking folic acid supplements, but only 16.6% took them at the recommended time. The main factors associated with low adherence to IFA supplementation included lack of knowledge about the importance of IFA supplementation, forgetfulness, and side effects. In Ethiopia, women who received information on IFA supplementation during antenatal care were more likely to adhere to IFA supplementation. In Italy, women with a higher education level and those who received information on the importance of folic acid supplementation were more likely to adhere to the recommended supplementation. In terms of consuming at least three months or 90 tablets, six out of ten respondents in our study met this compliance. This finding was in line with other studies conducted in India [38], Senegal [39,40], and Nepal [41]. Therefore, it is recognized that associated factors related to IFA compliance differ based on regional and economical status. 

Parental attainment, specifically maternal education, was a significant predictor of IFA consumption in this study. This result is similar to other studies [15,39,42,43,44]. Studies conducted in low-income countries suggest that maternal education plays a critical role in promoting adequate IFA intake among pregnant women. In Pakistan, a study showed that maternal academic background was the key predictor for consumption of IFA [5]. Similar findings were observed in a study from Ethiopia, where higher maternal education was associated with higher IFA intake [45]. The plausible reason behind this is that educated mothers are more likely to have learn about nutrition, increased health concerns, the importance of micronutrients, and ANC services in pregnancy and childbirth compared to their less educated peers [40,46]. Women with higher education levels are more likely to have greater decision-making power and autonomy in their households, which may enable them to prioritize their health and the health of their unborn child. Moreover, women with higher education levels may have better access to health information and services, which may result in increased awareness and adherence to IFA supplementation during pregnancy [47]. 

Advanced educational qualifications of the husband were also associated with IFA consumption for at least 90 days in our study. A growing body of evidence suggests that husband’s education level increases their engagement in reproductive health intervention, ultimately having a positive health outcome [48]. Educated husbands are probably more aware of the health benefit of supplementation, thus encouraging the consumption of IFA. However, why a similar effect was absent in consumption of 180 tablets is unclear and requires further study. 

In our analysis, receiving at least four ANC visits from medically skilled providers was a strong and significant predictor of higher consumption of IFA, regardless of consumption of either 90 tablets or 180 tablets. Other studies support our findings with similar results [42,43,49,50]. This is likely due to the fact that women who receive regular ANC checkups are more likely to be aware of the importance of IFA supplementation and are more likely to receive information and support from health care providers regarding the benefits of IFA. This may also indicate that frequent prenatal visits may play a central role in obtaining sufficient IFA, as it is provided to mothers during pregnancy by government health centers in Bangladesh. Adequate ANC visits could be the ultimate platform to communicate with health providers, increasing the dissemination of the health benefits of IFA, nutritional knowledge and practices among pregnant women. ANC services often provide a convenient opportunity for women to obtain IFA supplements, and women who attend four or more ANC visits are more likely to receive IFA supplements than those who attend fewer visits. 

The prevalence of intimate partner violence or IPV in this particular study was similar to other studies [51,52,53]. In addition to previously established predictors, we have found that IPV was a significant determinant of IFA consumption in terms of a minimum of WHO-recommended six months or 180 tablets. Although we did not find any significant association between IPV and consumption of at least 90 days of IFA in our study, respondents who faced intimate partner violence were 38% less likely to consume IFA for at least 180 days. Numerous previous studies reported domestic violence as a significant predictor of lower adherence to ANC, late entry to ANC, and inadequate utilization of maternal health services [24,25,54,55]. Therefore, we may assume a strong association between partner violence and IFA adherence. However, based on our findings, further investigation with robust design for exploring causal link should be carried out.

### Strengths, Limitations, and Further Scopes 

The major strength of our study is the representativeness and generalizability of the findings as we used a larger sample size of nationally representative data. Moreover, to balance the selection bias, non-responses, and other potential biases, sampling weights were used [32]. Conversely, this study has a few weaknesses related to the original dataset. Being a cross-sectional survey, we could not establish causality between outcome and independent variables. Furthermore, being self-reported data, there is a strong probability of recall bias, particularly attached to the outcome variable of number of tablets consumed. As we did not collect these data directly, we are not certain how these issues, such as recall bias or social desirability bias, were accounted for in the final dataset. Additionally, some variables that were found to be statistically significant to the consumption of IFA in studies from other countries, such as side effects, forgetfulness, stock out, unavailability of IFAS, the distance of health facilities and travel costs, knowledge of anemia, and history of safe delivery without consuming IFAS were unavailable in the BDHS dataset. Hence, our analyses did not account for these factors. Nevertheless, we encourage other researchers, government, donors, and non-governmental organizations to conduct advanced research and implement meaningful interventions to increase adherence to IFAS among pregnant women.

## 5. Conclusions

Only one in five pregnant women in Bangladesh consumes the WHO-recommended 180 iron–folic acid tablets during her pregnancy. Moreover, maternal age, education, ANC visits during pregnancy and IPV were also significant predictors of IFA consumption in terms of a minimum of WHO-recommended six months or 180 tablets. This paints a grim picture, as the Bangladesh government has invested significant effort to make this supplementation available everywhere without any cost. Based on the findings, we conclude that more precise context-specific intervention strategies must be developed and implemented with fidelity to overcome not only the established barriers but also to effectively counter future challenges.

## Figures and Tables

**Figure 1 nutrients-15-01512-f001:**
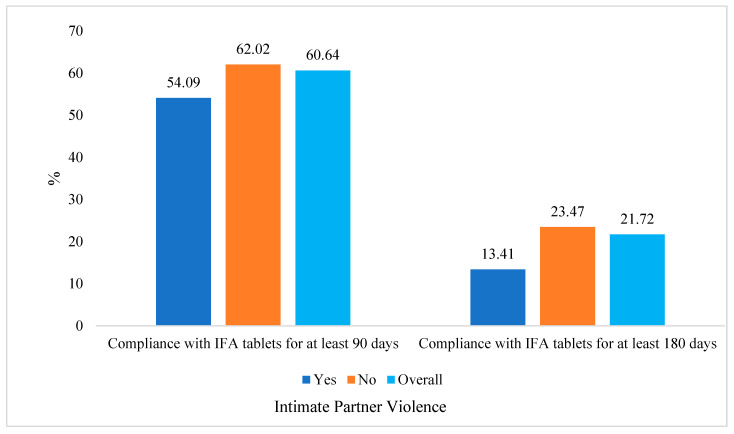
Distribution of iron–folic acid (IFA) compliance.

**Table 1 nutrients-15-01512-t001:** Background characteristics of the study population (*n* = 3828).

Indicators, *n* (Column %)	Urban (*n* = 1081)	Rural (*n* = 2746)	Total (3828)	*P*-Value
**Division**				
Barishal	36 (3.36)	157 (5.71)	193 (5.04)	<0.001
Chattogram	192 (17.75)	613 (22.33)	805 (21.04)	
Dhaka	508 (47.05)	478 (17.41)	987 (25.78)	
Khulna	86 (7.93)	277 (10.09)	363 (9.48)	
Mymensingh	56 (5.19)	260 (9.46)	316 (8.26)	
Rajshahi	93 (8.62)	365 (13.27)	458 (11.96)	
Rangpur	68 (6.27)	388 (14.12)	456 (11.9)	
Sylhet	41 (3.82)	209 (7.61)	250 (6.54)	
**Respondent’s age**				
15–19 years	183 (16.96)	532 (19.35)	715 (18.68)	0.026
20–34 years	827 (76.51)	2064 (75.12)	2890 (75.51)	
35–49 years	71 (6.53)	152 (5.53)	222 (5.81)	
**Respondent’s educational qualification**				
No education	53 (4.91)	126 (4.59)	179 (4.68)	<0.001
Primary	249 (23.08)	715 (26.02)	964 (25.19)	
Secondary	481 (44.54)	1427 (51.95)	1908 (49.85)	
Higher	297 (27.47)	479 (17.44)	776 (20.27)	
**Husband’s educational qualification**				
No education	94 (8.82)	336 (12.33)	430 (11.35)	<0.001
Primary	283 (26.66)	920 (33.78)	1204 (31.78)	
Secondary	372 (35.01)	958 (35.17)	1331 (35.13)	
Higher	314 (29.52)	510 (18.71)	824 (21.75)	
**Wealth Index of household**				
Poor	143 (13.27)	1295 (47.16)	1439 (37.59)	<0.001
Middle	124 (11.47)	610 (22.2)	734 (19.17)	
Rich	813 (75.26)	842 (30.64)	1655 (43.24)	
**Respondents currently working**	778 (72.01)	1639 (59.66)	2417 (63.15)	<0.001
**Household head’s sex was male**	982 (90.9)	2369 (86.25)	3352 (87.56)	<0.001
**At least four ANC from medically skilled providers ^1^**	696 (64.44)	1283 (46.71)	1980 (51.72)	<0.001
**Decision making by respondents ^2^**	636 (59.78)	1433 (52.6)	2069 (54.61)	<0.001
**Access to mass media**	893 (82.65)	1762 (64.16)	2656 (69.38)	<0.001
**Intimate partner violence**	155 (14.32)	511 (18.59)	665 (17.38)	<0.001

^1^ The skilled service providers considered in this study included MBBS doctors, nurse/midwife/paramedic, family welfare visitor, community skilled birth attendant, and sub-assistant community medical officer. ^2^ The items of decision making were major household purchases, health care, and visits to family or relatives.

**Table 2 nutrients-15-01512-t002:** Factors associated with iron–folic acid compliance for at least 90 days and 180 days.

Independent Variables	(a)At Least 90 Days		(b)At Least 180 Days	
	aOR (95% CI)	*P*-Value	aOR (95% CI)	*P*-Value
**Respondent’s age**				
15–19 years	Reference		Reference	
20–34 years	1.26 (1.03, 1.54)	0.026	1.11 (0.88, 1.41)	0.375
35–49 years	1.22 (0.85, 1.76)	0.274	1.51 (0.99, 2.31)	0.057
**Respondent’s educational qualification**				
No education	Reference		Reference	
Primary	1.11 (0.74, 1.68)	0.606	0.87 (0.48, 1.57)	0.638
Secondary	1.77 (1.16, 2.70)	0.008	1.63 (0.92, 2.91)	0.097
Higher	2.73 (1.65, 4.53)	<0.001	2.45 (1.34, 4.48)	0.004
**Husband’s educational qualification**				
No education	Reference		Reference	
Primary	1.30 (0.98, 1.72)	0.074	1.03 (0.70, 1.52)	0.879
Secondary	1.33 (1.00, 1.77)	0.049	1.08 (0.72, 1.62)	0.693
Higher	1.75 (1.22, 2.52)	0.002	1.36 (0.89, 2.10)	0.156
**Wealth Index of family**				
Poor	Reference		Reference	
Middle	0.84 (0.66, 1.06)	0.142	0.83 (0.61, 1.13)	0.236
Rich	0.98 (0.79, 1.21)	0.835	1.09 (0.84, 1.42)	0.495
**Currently working**				
Yes	Reference		Reference	
No	1.02 (0.85, 1.23)	0.798	1.24 (1.01, 1.52)	0.044
**At least four ANC from medically skilled providers**				
No	Reference		Reference	
Yes	2.53 (2.14, 3.00)	<0.001	2.43 (1.97, 3.00)	<0.001
**Access to Mass Media**				
No	Reference		Reference	
Yes	0.99 (0.82, 1.20)	0.937	0.83 (0.65, 1.05)	0.119
**Decision making by women**				
No	Reference		Reference	
Yes	1.16 (0.98, 1.38)	0.078	1.06 (0.87, 1.29)	0.574
**Intimate Partner Violence**				
No	Reference		Reference	
Yes	0.90 (0.74, 1.10)	0.313	0.62 (0.48, 0.81)	<0.001

All independent variables and sampling weight were included in the multiple logistic regression models where the outcome variables were (a) IFA compliance at least 90 days, and (b) IFA compliance at least 180 days.

## Data Availability

The DHS data is publicly available secondary data obtained from the 2017–2018 Bangladesh DHS. The data files are freely available from the DHS website (https://dhsprogram.com/data/available-datasets.cfm, accessed on 6 May 2022). Therefore, ethical approval is waived for future analysis. Access to the dataset was given upon email request for analysis. Ethical approval for the original Bangladesh DHS was obtained from the Inner City Fund (ICF) International institutional review board.

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
