# Peer review of "What Makes Bangladeshi Pregnant Women More Compliant to Iron–Folic Acid Supplementation: A Nationally Representative Cross-Sectional Survey Result"

_nutrients, 2023, doi:10.3390/nu15061512_

Round 1

Reviewer 1 Report

This is a cross-sectional study on compliance of iron-folic acid supplementation among Bangladeshi pregnant women. Although the goal of the study has some merit given the governmental efforts invested to provide free and easily accessible supplements to pregnant women, there are many problems including lack of precision and clarify in the manuscript that needs to be addressed before it can be considered for publication. Please see the specific comments below. The manuscript also needs to be revised by a professional revision service to improve writing style and correct mistakes.

Abstract

-          The ANC acronym is not defined.

-          Those below 90 days are not reported.

-          Instead of reporting only descriptive statistics, I would recommend reporting odds ratios for the most relevant and consistent factors (intimate partner violence was part of the main objective in the introduction of the study), antenatal visit and education were the strongest factors.  

Introduction

Lines 50-51: “Unfortunately, adherence to IFA during pregnancy is still lower in Bangladesh.”

-          Lower than what or which other populations?

Lines 54-5 “A study also conducted in three rural districts of Bangladesh revealed that only 18% of respondents consume IFA tablets during pregnancy [12].”

-          “consume IFA tables” is vague particularly given the fact that the average number of days is specified in the previous sentences. How does it compare to other studies?

Methods

Lines 85-86: “final (weighted) sample size was 3,828.

-          Why is “weighted” between brackets? What was weighted? Please clarify.

-          Line 105 alludes to it but is still unclear. Please define how the sample was weighted.

Line 88: iron and syrup intake was reported. Did you analyze whether the factors were the same for both? It may be possible that one is more convenient than the other or that side effects differ and may affect  compliance. Have you compared them? This may be informative for healthcare providers and policy-makers.

Line 94: The outcome variables were at least 90 days of intake or 180 days (3 and 6 months) based on other studies and on WHO recommendations. Since the number of days was reported by all women in the final dataset, it is possible to look at the determinants of compliance as a continuous outcome variable. The 3 or 6 months could have been taken at any time point during the pregnancy or intermittently which may affect iron status. Since you have the full spectrum of intake during pregnancy, it would be worth taking advantage of the richness of the data and see if the determinants are consistent with the somewhat arbitrary cut points. There is no need to show all of these analyses but could be mentioned in the text if comparable or different when examined as a continuous outcome.

Lines 97-102: Independent variables – the description of independent variables is too vague. It is not possible to critically appraise the assessment or classification of those variables without more details. For examples, what is the wealth index of family (how is it calculated, is it a validated score?), why chose at least 4 ANC and not use it as an ordinal or continuous variable to see how many visits are actually associated with optimal compliance, what does “decision-making by respondents” mean, same for mass media – how was it measured and finally intimate partner violence should be better defined since it is part of your main objective.

Statistical analysis

– lines 103-114: Please specify how the sample was weighted.

-          A X2 test was used to identify the association between the independent and dependent variables. It is not possible to determine if this test is appropriate since we cannot know by the description above whether the independent variables are continuous or categorical (without having to refer to the tables). Better describing the independent variables and its categorization would solve this issue.

-          How was the analysis of confounders performed, which confounders were retained and why?

-          Lines 109-111: “After adjusting the relevant co-variates, a multiple logistic regression analysis was used to estimate the adjusted odds ratios (ORs) as the strength of association between compliance with IFA tablets AT 90 DAYS AND 180 DAYS and its associated factors. » Please add the two main outcomes as suggested in capitalized and underlined letters.

Results

Lines 116-121: Since those regions of Bangladesh are not known to most readers from other countries, it would be beneficial to specify whether they are rural or urban divisions.

Lines 128-129: “The lowest number of pregnant mothers from low-income families met the doses of a minimum of 90 days and 180 days.” – This sentence is unclear. Please clarify.

Line 132 – please provide the % among working women to help the reader to compare with housewives.

Line 132: “More than half of the women (54.61%) could take the decision.” – Please clarify what decision is discussed in this sentence (decision of taking a supplement?). Line 134 – what do you mean by “without decision-making capability? Because of mental health disabilities or because their husband takes the decision for them?

Table 1 is cluttered. I recommend reporting only the “yes” column as the “no” column is automatically the remaining of the sub-group (of 90 or 180 days). Also, instead of writing p=0.000, please put p<0.001. Why is there a “+” sign after “Decision making respondents+”? There is no footnote at the bottom of the table to explain it.

Lines 145-153: OR values are repeated 3 times; ex. 1.26 times followed by the actual AOR and CI and then repeated in the table. You can mention the AOR and refer to the table for complete values. Same comment for the following paragraph.

AOR stands for adjusted odds ratio but to my knowledge, the variables used for the adjustment are not specified anywhere. Please specify in the methods and results (ex. table footnote).

Table 2 – there is a “b” at “At least 4 ANC b” …but this is not specified in the table footnote.

Figure 1 shows “at lease 60 days” as a new cut point. This may be of interest but has not been defined nor justified in the methods section and data for 60 days are not shown elsewhere. Please standardize the methods and results throughout the manuscript. Perhaps simply remove it in this figure to avoid overloading other tables.

Please spell out the acronyms in table and figure titles. Is there a statistical difference between groups in this figure 1? Please specify.

It turns out that having 4 antenatal visits is a strong predictor of compliance. This cut point of 4 visits seems to have been chosen somewhat arbitrarily. Since this factor is a strong predictor of the outcome, it would be worth examining this data in greater depth to see if there is a minimum threshold or optimal number of visits (if the number of visits is available in the dataset).

Discussion

Many explanations for the role of each factor on compliance are speculative and are not supported by references. Also, the authors mention that other studies found similar (or differing) results without giving any details on those other studies (comparable populations, methods, outcomes?). These issues should be addressed.

Line 214 – Paternal attainment should be replaced by ‘Parental” since maternal education is mentioned in the same line.

Lines 223-223: “A study conducted in India found that the presence of husbands at the time of ANC was 223 positively associated with IFA compliance [30].” This sentence is not relevant to paternal education so should be removed or moved elsewhere if relevant.

Line 226 – Please provide more details about the reference 30. Details about the study might explain part of the discrepancies between studies.

Lines 237-239: “direct link between partner violence and IFA adherence”- you cannot infer causality with an observational study design. Please remove the sentence or modify it.

The only weakness raised in the paragraph (p.8), apart from being cross-sectional, is about the data that was not collected. Other weaknesses could be brought up including the way data was collected (ex. self-reported) and potential recall or social desirability bias which may have affected the assessment of the outcome (compliance).  

Conclusion

Very good conclusion.

Author Response

We sincerely appreciate your kind and thoughtful review and suggestions, these were really helpful in improving the quality of the manuscript further. We are providing a point by point response as suggested in the attached file.

Reviewer 2 Report

The study examined what factors determining the compliance to iron-folic acid supplementation in pregnant women from Bangladesh. The topic is very important, especially in this population. The results confirm the importance of nutritional counseling in this area. An interesting observation is the impact of experiencing partner violence on the use of nutritional intervention, especially its duration. These data are shown in Figuret 1, so my question is whether a statistical analysis (Chi square test) was performed for these data?

In addition, it was shown that the mother's age was also a factor that determined compliance to supplementation. Do the authors know wheter these were the first or subsequent pregnancies for the women?

Author Response

We sincerely appreciate your kind and thoughtful review and suggestions, these were really helpful in improving the quality of the manuscript further. We have revised the whole manuscript
